# A Method for Sorting High-Quality Fresh Sichuan Pepper Based on a Multi-Domain Multi-Scale Feature Fusion Algorithm

**DOI:** 10.3390/foods13172776

**Published:** 2024-08-30

**Authors:** Pengjun Xiang, Fei Pan, Xuliang Duan, Daizhuang Yang, Mengdie Hu, Dawei He, Xiaoyu Zhao, Fang Huang

**Affiliations:** 1College of Information Engineering, Sichuan Agricultural University, Ya’an 625014, China; 2022219014@stu.sicau.edu.cn (P.X.); fei.pan@sicau.edu.cn (F.P.);; 2Ya’an Digital Agricultural Engineering Technology Research Center, Ya’an 625014, China

**Keywords:** Sichuan pepper sorting, instance segmentation, maturity classification, machine vision, smart agriculture

## Abstract

Post-harvest selection of high-quality Sichuan pepper is a critical step in the production process. To achieve this, a visual system needs to analyze Sichuan pepper with varying postures and maturity levels. To quickly and accurately sort high-quality fresh Sichuan pepper, this study proposes a multi-scale frequency domain feature fusion module (MSF3M) and a multi-scale dual-domain feature fusion module (MS-DFFM) to construct a multi-scale, multi-domain fusion algorithm for feature fusion of Sichuan pepper images. The MultiDomain YOLOv8 Model network is then built to segment and classify the target Sichuan pepper, distinguishing the maturity level of individual Sichuan peppercorns. A selection method based on the average local pixel value difference is proposed for sorting high-quality fresh Sichuan pepper. Experimental results show that the MultiDomain YOLOv8-seg achieves an $mAP_{50}$ of 88.8% for the segmentation of fresh Sichuan pepper, with a model size of only 5.84 MB. The MultiDomain YOLOv8-cls excels in Sichuan pepper maturity classification, with an accuracy of 98.34%. Compared to the YOLOv8 baseline model, the MultiDomain YOLOv8 model offers higher accuracy and a more lightweight structure, making it highly effective in reducing misjudgments and enhancing post-harvest processing efficiency in agricultural applications, ultimately increasing producer profits.

## 1. Introduction

Sichuan pepper (Zanthoxylum bungeanum) is a spice and traditional Chinese medicine originating from Sichuan Province, China, renowned for its unique flavor and medicinal value [1,2,3]. China is the largest producer of Sichuan pepper globally, making it an important specialty crop with significant economic value [4]. The quality of Sichuan pepper can be influenced by environmental factors such as the growing region, climate, and temperature, leading to varying quality levels [5]. Temperature changes can affect the ripening of Sichuan peppers; under high temperatures, the fruit may ripen too quickly, affecting its flavor and aroma. Excessive rainfall increases the rate of fruit cracking, thereby reducing the quality and market value of the Sichuan peppers. Therefore, grading fresh Sichuan peppercorns can meet the diverse needs of different markets, better ensuring quality control and enabling producers to maximize returns and resource utilization. Grading fresh Sichuan peppercorns also improves the efficiency and quality of subsequent secondary processing, which has a significant economic impact. Adopting different processing methods according to the quality of the Sichuan peppercorns can reduce post-harvest losses, decrease the rate of defective products, and improve overall product quality. The enhancement in efficiency and quality directly relates to increased profitability for producers. Graded Sichuan peppercorns are better positioned to meet the varying demands of the market, promote product diversification, and enhance added value.

To promote intelligent, sustainable, and efficient agricultural development, smart agriculture has become a key direction for future agricultural development. Agricultural robots are essential for processes such as planting, irrigation, and harvesting, with machine vision being the core component of these robots [6]. With the rapid advancement of vision technology, deep learning-based vision segmentation techniques are widely used in agriculture [7,8,9]. Agricultural robots use visual systems to analyze crops from multiple angles and dimensions, performing tasks such as sorting, harvesting, and picking. However, cluster crops like Sichuan pepper, grapes, and lychees pose challenges for agricultural machine vision systems due to their clustered growth, overlapping, and complex backgrounds [10,11]. Peng et al. [12] constructed the RDf-DeepLabV3+ network to enhance the segmentation performance of lychees, achieving a mIoU of 77% under complex background conditions. Wang et al. [13] designed the DualSeg network, based on CNN and transformer, for the segmentation of grape clusters and branches, achieving an mIoU of 82.7%. Xu et al. [14] proposed an improved Mask R-CNN network for the visual system of agricultural robots, achieving a recognition accuracy of 93.76% for cherry tomatoes. Wang et al. [15] developed a perception model for the detection and segmentation of Sichuan pepper and branches, achieving an $mAP_{50}$ of 90.1% with an mIoU of 84.2% and 89.9% for Sichuan pepper and branches, respectively, aiding in pepper harvesting. The above studies focus only on the identification and segmentation of crops during the growth process, without exploring the post-harvest handling of crops. This research will further investigate the post-harvest processing of Sichuan peppers and analyze their quality to support the subsequent selection of high-quality Sichuan peppercorns.

To enhance crop economic value, it is crucial to accurately grade crop quality quickly after analysis. Machine vision plays a vital role in crop grading [16,17]. Ji et al. [18] proposed a multi-view method for grading apple quality, using an improved YOLOv5s algorithm to detect and grade defects from multi-angle images. Deng et al. [19] developed an online carrot grading system based on machine vision and deep learning, using the CDDNet network to detect defects and MBR fitting and convex polygon approximation for quality grading. TU et al. [20] used image recognition software and machine learning to separate high-quality pepper seeds from inferior ones, using eight features such as color, width, length, projection area, and single-core density for prediction with a multilayer perceptron. Wang et al. [21] graded dried Sichuan pepper using image processing and BP neural networks, achieving a classification accuracy of 79.53% to 98.04% by extracting RGB and HIS values from pepper images. Although deep learning performs well in image processing, it also introduces challenges such as efficiency issues, complex feature extraction, high computational complexity, and a large number of network parameters. On the other hand, while machine learning algorithms offer good processing speed, they perform poorly with high-dimensional data-like images [22]. Moreover, they are less effective than deep learning in handling large-scale data, making it difficult to meet the real-time grading requirements for crop quality.

In actual sorting processes, model performance and multi-dimensional image analysis are necessary to ensure sorting algorithm accuracy. Currently, there is a lack of research on quality grading for fresh Sichuan pepper. Traditional manual sorting is inefficient, inaccurate, and labor intensive. As a specialty economic crop with high economic and medicinal value, integrating emerging computer and AI technologies into the Sichuan pepper industry can better support its development. Therefore, developing an algorithm for sorting high-quality Sichuan pepper to efficiently, quickly, and accurately grade its quality is essential. This paper addresses the challenges of deep learning networks’ computational load, high parameter count, and complex feature extraction by enhancing model accuracy with spatial, frequency, and channel domain feature information. Additionally, the network is lightweight to suit real-time sorting requirements in agricultural scenarios, proposing a new algorithm for sorting high-quality fresh Sichuan pepper. The main contributions of this paper are as follows:Proposed a multi-scale frequency domain feature fusion module (MSF3M) to capture multi-scale frequency domain image features.Proposed a multi-scale dual domain feature fusion module (MS-DFFM) to capture multi-scale channel and spatial domain features.Developed a high-quality fresh Sichuan pepper sorting algorithm based on the average local pixel value difference method.

The rest of the paper is organized as follows: Section 2 details the methods for Sichuan pepper dataset collection and construction, the MultiDomain YOLOv8 model network, the MSF3M and MS-DFFM module construction, and the high-quality fresh Sichuan pepper sorting algorithm implementation. Section 3 presents extensive comparative and ablation experiments to evaluate the proposed approach, demonstrating its scientific validity and rationality. Section 4 discusses the limitations of the proposed algorithm. Section 5 provides a general summary of the study.

## 2. Materials and Methods

### 2.1. Overview of the Proposed Fresh Sichuan Pepper Quality Classification System

The overall process of the proposed fresh Sichuan pepper quality classification system is illustrated in Figure 1, consisting of three main stages.

In the first stage, as shown in Figure 1b, images of mature and semi-mature Sichuan peppercorns are segmented and labeled, as depicted in Figure 1a. The MultiDomain-YOLOv8-seg model is used to learn the features of individual peppercorns.

In the second stage, as shown in Figure 1d, images of pepper clusters from the front and back views are collected, as illustrated in Figure 1c. The best-trained MultiDomain-YOLOv8-seg model is then applied for prediction, and the coordinates of the predicted peppercorns are used to crop single peppercorn images from the original images, as shown in Figure 1e.

In the third stage, as shown in Figure 1f, the segmented dataset is first input into the MultiDomain-YOLOv8-cls classification network for feature learning, aiming to classify mature and semi-mature peppercorns. Next, the average local pixel value difference of the predicted mature peppercorns is calculated to verify the color uniformity. Finally, by analyzing the sparsity and proportion of black pixels in the images, high-quality peppercorns are selected for future classification of fresh Sichuan pepper quality. The specific algorithm flow is represented in the generated image, as shown in Figure 1g.

### 2.2. Data Acquisition and Annotation

To better train the model, we collected a dataset of Sichuan peppers with varying degrees of ripeness in Hanyuan County, Sichuan Province, China. To meet the grading requirements, we mainly collected semi-mature and mature fresh Sichuan peppers from Hanyuan and labeled the collected data according to the segmentation task requirements. The peppers were photographed during both morning and afternoon sessions on a single day. Since the ripeness of the Sichuan peppers is influenced by altitude and temperature, we captured images of peppers with different ripeness levels from both the mountaintop and the foothills at the Hanyuan Sichuan pepper plantation. A standard smartphone camera was used to capture the images.

During the shooting process, individual pepper clusters were photographed from a front-lit direction at a distance of approximately 5–10 cm, using consistent and stable artificial lighting to avoid the impact of natural light on image quality. A total of 1942 images were collected and labeled. The dataset was split in a 9:1 ratio, with 1624 images as the training set, 181 as the validation set, and 100 images selected as the test set to evaluate the segmentation performance of the proposed algorithm.

The collected Sichuan pepper images were labeled using the AnyLabeling tool, which integrates the segment anything model (SAM). This open-source labeling platform supports multiple model integrations and can automatically generate precise annotation data. The Sichuan peppers in the images were labeled at the granule level, and the corresponding collected images and labeling results are shown in Figure 2.

To validate the performance of the high-quality Sichuan pepper classification algorithm proposed in this study, we collected 520 images of post-harvest Sichuan pepper clusters, including 246 clusters of semi-mature peppers and 274 clusters of mature peppers. Due to the influence of environmental factors such as light, rainfall, and temperature, there are significant differences in the ripeness of the front and back sides of the Sichuan peppers. These differences may affect the model’s ability to accurately assess the quality of the peppers. To capture these potential variations, images of both the front and back sides of the pepper clusters were taken during the data collection process. This approach allows the model to acquire feature information from different angles, leading to more accurate identification and classification of the ripeness and quality of the peppers.

This multi-angle data collection strategy not only increases the diversity of the dataset but also enhances the model’s classification accuracy and generalization ability, ensuring the effective selection of high-quality Sichuan peppers. During the data collection process, all images were captured under the same conditions and lighting setup. The corresponding collected images are shown in Figure 3.

### 2.3. Sichuan Pepper Instance Segmentation Model: MultiDomain YOLOv8-seg

This section provides a detailed explanation of the proposed Sichuan pepper instance segmentation model, MultiDomain YOLOv8-seg. YOLOv8 is flexibly designed and easy to integrate, offering outstanding segmentation and classification performance. It maintains high accuracy while achieving faster inference speed.

The segmentation model is designed based on the YOLOv8 segmentation framework, introducing a multi-scale frequency domain feature fusion module (MSF3M) and a multi-scale dual-domain feature extraction module (MS-D2F2M) to integrate multi-domain and multi-scale image feature information. The model combines feature extraction in the frequency domain, spatial domain, and channel domain, leveraging the advantages of all three to enhance the model’s feature representation capability and robustness. The specific design includes key components such as an attention module, multi-scale feature information fusion, and an up-sampling module. It also incorporates programmable gradient information to achieve a lightweight deep neural network with high accuracy and fast inference speed.

In the backbone network, the RepNCSPELAN4 and Adown modules from YOLOv9 [23] are introduced. The architecture of the RepNCSPELAN4 module is shown in Figure 4, which is used to reduce the network parameter count while improving accuracy. In the shallow network, where the receptive field is small, it can extract low-level feature information such as edges, textures, and colors. The RepNCSPELAN4 module efficiently aggregates multi-scale, multi-layer detailed feature information in the shallow network. As the network depth increases, semantic feature information becomes richer, resulting in a significant increase in computational load. Therefore, using RepNCSPELAN4 would cause a substantial increase in computation and parameters. To address this, the C2f module with an integrated SE attention mechanism [24] is used to focus on key feature information and accelerate network inference operations. Additionally, a multi-scale frequency domain feature fusion module (MSF3M) is proposed in the deep network for extracting multi-scale frequency domain feature information, capturing different frequency components in the input image to enhance the model’s feature extraction performance and accuracy. Finally, the SPPELAN module replaces the SPPF module in YOLOv8, combining the advantages of SPP and ELAN, further enhancing feature extraction capability through an efficient layer aggregation strategy.

In the neck part, traditional spatial up-sampling operators rely heavily on local pixel focus and fail to explore global dependencies. Therefore, a deep Fourier up-sampling operator [25] is introduced to address the issue of CNN networks in deep networks being unable to capture long-distance feature information. Additionally, a multi-scale dual-domain feature fusion module (MS-D2F2M) is proposed, which extracts feature information from different scales in the spatial and channel domains to improve network accuracy. The proposed network integrates multi-scale spatial, frequency, and channel domain features, performs feature fusion processing, and introduces lightweight modules and attention modules to further accelerate network inference, significantly enhancing segmentation accuracy and reducing network computational complexity.

### 2.4. Multi-Scale Frequency Domain Feature Fusion Module (MSF3M)

This section provides a detailed explanation of the multi-scale frequency domain feature fusion module (MSF3M), hereafter referred to as MSF3M. Its structure is shown in Figure 5. This module is designed for multi-scale frequency domain feature extraction in deep networks. Deep networks often face challenges such as gradient vanishing, overfitting, and high computational complexity during image feature extraction [26]. Frequency domain feature extraction offers advantages over spatial domain feature extraction in terms of global information capture, translation invariance, and multi-scale feature fusion. Therefore, the MSF3M module is proposed for multi-scale feature extraction in deep networks. The MSF3M module consists of four main parts:

Part One: In the first part of the MSF3M module, the input image undergoes a fast Fourier transform (FFT) to convert the image from the spatial domain to the frequency domain, resulting in a complex representation with magnitude and phase components. The magnitude primarily describes the intensity of frequency components, while the phase describes the phase shift of frequency components. Phase information plays a crucial role in image feature extraction and reconstruction, while magnitude information contains most of the image’s texture and structure.

Part Two: In the second part of the MSF3M module, multi-scale feature extraction is performed on the magnitude and phase information. The feature extraction process for magnitude and phase is consistent; this part uses magnitude as an example. First, the magnitude (Magnitude) undergoes feature extraction using a 5 × 5 convolution. Then, the feature-extracted image is subjected to three sets of layered convolutions: 1 × 7 and 7 × 1, 1 × 11 and 11 × 1, and 1 × 21 and 21 × 1. This results in multi-scale feature information for the magnitude. Finally, a 1 × 1 convolution layer is applied to increase non-linearity and achieve cross-channel information fusion, thereby obtaining more feature information.

Part Three: In the third part of the MSF3M module, the multi-scale convolutions’ outputs, Mag_att (magnitude) and Pha_att (phase), are used for frequency domain reconstruction. The real part of the complex matrix is obtained by multiplying Mag_att by the cosine of Pha_att. Similarly, the imaginary part of the complex matrix is obtained by multiplying Mag_att by the sine of Pha_att. The real and imaginary parts are then combined to form the complex matrix in the frequency domain.

Part Four: In the fourth part of the MSF3M module, the complex matrix obtained from the third part undergoes an inverse fast Fourier transform (IFFT) to convert the image back from the frequency domain to the spatial domain. The output is then multiplied by the original input to retain important features from the original input image, thereby enhancing the model’s performance.

#### 2.4.1. Multi-Scale Frequency Domain Feature Fusion Module Part I

MSF3M Part One: After applying the fast Fourier transform (FFT) to the input image, the image is transformed from the spatial domain to the frequency domain. The frequency domain image represents the image information in the frequency domain, where the low-frequency components are located in the central region of the frequency domain image, preserving the overall structure and primary feature information of the image. The high-frequency components are located at the edges of the frequency domain image, preserving the details and edge information of the image. Furthermore, the magnitude and phase of the frequency domain signal are extracted. The magnitude retains the main feature structure of the image, while the phase retains the spatial information of the image. Therefore, multi-scale feature extraction is subsequently performed on the magnitude and phase information.

In this study, the FFT algorithm is used to transform the input features from the spatial domain to the frequency domain. The fast Fourier transform is an efficient algorithm for computing the discrete Fourier transform (DFT). It decomposes an N-point DFT into two $\frac{N}{2}$-point DFTs, using a recursive approach for computation. The time complexity of the FFT is only $O\left(N\log N\right)$, significantly reducing the computational load compared to the $O\left(N^{2}\right)$ complexity of the DFT, thus accelerating network inference and analysis. The corresponding fast Fourier transform formula can be expressed as:(1)$$\hat{X}\left(k,l\right)=\sum_{x=0}^{M-1}\sum_{y=0}^{N-1}X\left(x,y\right)\cdot e^{-j2\pi \left(\frac{kx}{M}+\frac{ly}{N}\right)}$$
where $\hat{X}\left(k,l\right)$ represents the feature map in the frequency domain, $X\left(x,y\right)$ represents the feature map in the spatial domain, j is the imaginary unit, N and M denote the dimensions of the input signal.

To obtain the magnitude and phase of the frequency domain feature information, the complex matrix in the frequency domain is further processed. Using Euler’s formula, the real part matrix $Re\left(\hat{X}\left(k,l\right)\right)$ and the imaginary part matrix $Im\left(\hat{X}\left(k,l\right)\right)$ are derived from the complex matrix. Furthermore, the formulas for the magnitude (2) and phase (3) in the frequency domain can be expressed as:(2)$$Magnitude\left(\hat{X}\left(k,l\right)\right)=\sqrt{{Re\left(X\left(k,l\right)\right)}^{2}+{Im\left(X\left(k,l\right)\right)}^{2}}$$
(3)$$Phase(\hat{X}\left(k,l\right))=arctan(\frac{Im\left(\hat{X}\left(k,l\right)\right)}{Re\left(\hat{X}\left(k,l\right)\right)})$$
where $Magnitude\left(\hat{X}\left(k,l\right)\right)$ denotes the magnitude, and $Phase(\hat{X}\left(k,l\right))$ denotes the phase. The transformation from the spatial domain to the frequency domain, including the magnitude image and phase image, is illustrated in Figure 6.

#### 2.4.2. Multi-Scale Frequency Domain Feature Fusion Module Part II

MSF3M Part Two: Multi-scale feature extraction is performed on the magnitude and phase information in the frequency domain image to further obtain the image’s feature information.

To further extract features from the magnitude and phase, a 5 × 5 convolution operation is first applied to both the magnitude and phase to extract overall features. The resulting images are then processed through three sets of layered convolutions with different kernel sizes: 1 × 7 and 7 × 1, 1 × 11 and 11 × 1, and 1 × 21 and 21 × 1, to capture features at different scales. After feature extraction, the multi-scale feature information is aggregated. Finally, a 1 × 1 convolution is applied to reduce dimensionality, decreasing computational costs, and enhancing the network’s expressive capability. The final outputs are the multi-scale feature-extracted magnitude information, Mag_attn, and phase information, Pha_att. The formula for the multi-scale convolution can be expressed as:(4)$$(x\times h)(i,j)=\sum_{m}\sum_{n}x(i-m,j-n)\cdot h(m,n)$$
where x represents the input image features and h represents the convolution kernel.

#### 2.4.3. Multi-Scale Frequency Domain Feature Fusion Module Part III

MSF3M Part Three: Reconstruction of the frequency domain feature information involves reconstructing the magnitude information, Mag_attn, and phase information, Pha_att, into a complex matrix in the frequency domain.

The magnitude information, Mag_attn, and phase information, Pha_att, obtained after multi-scale feature extraction are reconstructed into the real part, $Re\left(X\left(x,y\right)\right)$, and the imaginary part, $Im\left(X\left(x,y\right)\right)$, of the complex matrix. The relevant calculation formulas are expressed as:(5)$$Re\left(X\left(x,y\right)\right)=Mag\_attn\cdot \cos\left(Pha\_att\right)$$
(6)$$Im\left(X\left(x,y\right)\right)=Mag\_attn\cdot \sin\left(Pha\_att\right)$$

After obtaining the real and imaginary parts of the complex matrix, the real part matrix and the imaginary part matrix are combined to form the complex matrix C, as represented below:(7)$$C=Re\left(X\left(x,y\right)\right)+j\cdot Im\left(X\left(x,y\right)\right)$$

#### 2.4.4. Multi-Scale Frequency Domain Feature Fusion Module Part IV

MSF3M Part Four: The frequency domain feature map is transformed back to the spatial domain using the inverse fast Fourier transform (IFFT). The spatial domain image is then multiplied with the original input feature map to further enhance the image’s feature information.

To transform the frequency domain feature map back to the spatial domain, the inverse fast Fourier transform is applied. This process can be represented as:(8)$$X(x,y)=\frac{1}{NM}\sum_{k=0}^{N-1}\sum_{l=0}^{M-1}X(k,l)\cdot e^{j2\pi (\frac{kn}{N}+\frac{lm}{M})}$$

At this stage, the multi-scale feature information in the frequency domain has been extracted. However, transforming the frequency domain image back to the spatial domain may introduce information loss or errors. To mitigate this, further feature enhancement is performed by multiplying the original spatial domain image with the transformed image. This operation preserves the image’s feature information to a large extent, resulting in an enhanced feature map. The corresponding calculation can be expressed as:(9)OUTPUT=X(x,y)×INPUT

As illustrated in Figure 7, the feature heatmaps after applying the MSF3M module exhibit more uniform activation areas, and more dispersed feature information, and retain more detailed information with less noise, resulting in higher-quality feature extraction from the images.

### 2.5. Multi-Scale Dual-Domain Feature Fusion Module (MS-DFFM)

This section introduces the multi-scale dual-domain feature fusion module (MS-DFFM), hereafter referred to as MS-DFFM. Its architecture is shown in Figure 8. This module integrates channel feature information and multi-scale spatial feature information, ultimately outputting image features enhanced by dual attention mechanisms. The module consists of two main parts: the channel attention mechanism and the multi-scale attention mechanism.

First, the SE (squeeze-and-excitation) channel attention mechanism is used to extract channel feature information from the image. The input tensor is first passed through an adaptive pooling layer and a 1 × 1 convolution. Then, it undergoes a nonlinear transformation via the HSigmoid activation function. The resulting output is multiplied by the input tensor to form the channel attention mechanism.

Next, the vector containing channel feature information is fed into the multi-scale convolution attention module [27], propagating along the multi-scale spatial dimensions. Initially, it goes through a 1 × 1 convolution for feature transformation, followed by a nonlinear transformation using the GELU activation function. Meanwhile, the input feature data enters the multi-scale convolution attention mechanism for multi-scale spatial feature extraction. This begins with a 5 × 5 convolution for feature transformation, followed by feature extraction using four different scales of convolution: 1 × 3 and 3 × 1, 1 × 7 and 7 × 1, 1 × 11 and 11 × 1, and 1 × 21 and 21 × 1. Finally, the multi-scale feature information is dimensionally reduced through a 1 × 1 convolution. This reduced feature map is then added to the original input via a residual connection, resulting in features enhanced by dual attention mechanisms. This enhancement further increases the network’s accuracy and performance.

### 2.6. Deep Fourier Up-Sampling Module

Traditional up-sampling operators often heavily rely on local pixel attention [28] and typically use fixed interpolation strategies, making it difficult to adjust strategies based on the input image content. They also lack the ability to model contextual information, which can result in poor performance for certain tasks such as super-resolution reconstruction and image generation. Deep neural networks, particularly convolutional neural networks (CNNs), perform convolutions locally, operating only on local input data and lacking the ability to extract contextual information. Even though methods like varying convolution kernel sizes and pyramid pooling are used to capture multi-scale features, they struggle to address the issue of up-sampling operators relying on local pixels.

In this study, a deep Fourier up-sampling module is introduced to model global characteristics and extract long-range dependencies. After applying the fast Fourier transform (FFT), the frequency domain information exhibits global properties. Processing this frequency domain information is more suitable for extracting global characteristic information.

The deep Fourier up-sampling in this study addresses the up-sampling problem by cyclically padding the amplitude and phase of the frequency domain signals. This module uses the FFT to obtain the amplitude and phase components of the frequency domain and then cyclically pads these two components in the H and W dimensions. Feature extraction is performed using a convolution module, and finally, the image is transformed back to the spatial domain using the inverse fast Fourier transform (IFFT). The padding process is illustrated in Figure 9.

### 2.7. A Method for Quality Sorting of Fresh Sichuan Pepper

This section will discuss the quality sorting of freshly picked Sichuan peppercorns, primarily using visual assessment to preliminarily grade the color quality of the peppercorns. Table 1 shows the quality grading standards for Hanyuan Sichuan peppercorns [29].

In this study, the quality sorting of Sichuan pepper is primarily based on the color of the peppers, focusing on two aspects: the ripeness of the peppers and the color uniformity of individual pepper grains. Detailed explanations of these two aspects are provided below.

To effectively classify mature and semi-mature Sichuan peppers after harvesting, we photographed the harvested peppers in the same scene. Due to various environmental factors during growth, peppers may not ripen uniformly. Therefore, we photographed both the front and back of the pepper clusters to ensure classification accuracy. The images were then input into the MultiDomain YOLOv8-seg model for segmentation, and the predicted segmentation coordinates were used to crop the original images. The cropped individual pepper grains were saved. Based on their appearance, the peppers were saved into folders for mature and semi-mature peppers. The dataset includes 2246 images of mature peppers and 1962 images of semi-mature peppers, which were divided into training and testing sets in an 8:2 ratio, as shown in Figure 10.

As shown in Figure 10, there are significant differences in ripeness and color between the front and back of the peppers. Therefore, assessing the quality from a single angle would be inaccurate. This study measures the peppers from two angles to validate the proposed algorithm’s reasonableness and accuracy.

### 2.8. Sichuan Pepper Classification Model: MultiDomain YOLOv8-cls

To better validate the performance of the proposed multi-scale frequency domain feature fusion module (MSF3M) and multi-scale dual domain feature fusion module (MS-DFFM), and to rapidly classify mature and semi-mature Sichuan peppercorns, this study proposes the MultiDomain YOLOv8-cls model. This model is designed based on the YOLOv8n-cls model, with the overall design structure shown in Figure 11.

In the network structure design, the MS-DFFM module is added to the shallow layers to extract multi-scale channel and spatial feature information. The MSF3M module is integrated into the deeper layers to focus on long-range dependencies by extracting multi-scale frequency domain features, retaining the rich feature information of the image. This significantly enhances the network’s accuracy and performance. The proposed network is lightweight, with low computational complexity and a small number of parameters, making it highly suitable for cost-sensitive smart agriculture scenarios where memory and computational resources are limited. The proposed classification network can better fit the requirements of smart agriculture, promoting agricultural intelligence. 

#### High-Quality Fresh Sichuan Peppercorn Sorting Based on Average Local Pixel Value Difference

In this section, we will elaborate on the quality assessment and sorting process for mature Sichuan peppercorns post-classification. High-quality fresh peppercorns should have characteristics such as full, vibrant colors and uniformity. We propose a high-quality peppercorn sorting algorithm based on the average local pixel value difference. This algorithm consists of two main parts:

Average Local Pixel Value Difference Method: This method determines regions in the image where the color or grayscale values change minimally by calculating the average local pixel value difference. Regions with small differences exhibit relatively uniform color or grayscale without distinct boundaries or textures. This method effectively distinguishes the flat color areas in peppercorn images, further determining color uniformity. The algorithm flow is illustrated in Figure 12.

The algorithm is divided into four parts:(1)Image Reading and Grayscale Conversion: Read the peppercorn image and convert it to a grayscale image.(2)Application of Convolution Filter: Calculate the average value of each pixel and its neighboring pixels using a 3 × 3 convolution kernel, and then compute the difference between each pixel and the average value of its neighbors (vALD).(3)Calculation of Global Average Pixel Value and Difference: Compute the global average pixel value of the image and the difference between each pixel and the global average.(4)Flat Region Marking: Mark the flat color regions of the image based on the previous calculations. Pixels with a difference lower than 10% of the global average pixel difference are marked white to produce the flat region labeled image.

Flat Region Labeled Image Assessment: To determine the color uniformity of the peppercorns, the flat region labeled images generated by the average local pixel value difference method are further classified. This assessment involves evaluating the sparsity and proportion of black pixels in the image, as detailed in Figure 13.

This algorithm is divided into three parts:(1)Black Pixel Proportion Calculation: Calculate the proportion of black pixels in the entire image, representing the proportion of non-flat regions.(2)Local Sparsity Detection: In the generated flat region labeled image, black pixels might be scattered throughout. If the proportion of black pixels in the entire image is low, the image is also considered non-flat. A sliding window is used to calculate the black pixel proportion in each window.(3)Global Sparsity Detection: Calculate the global black pixel proportion and compare it with a given global threshold. Images with a black pixel proportion lower than the threshold are classified as high-quality peppercorns.

By utilizing these methods, we can effectively screen and identify high-quality Sichuan peppercorns based on their visual characteristics, particularly focusing on color uniformity and maturity.

### 2.9. Implementation Details

This section details the experimental procedures for both the MultiDomain YOLOv8-seg and MultiDomain YOLOv8-cls models. No pre-trained weights were used during model training, as this study introduces significant modifications to the YOLOv8 model, including adjustments to the network structure and modules, leading to notable differences from the original model. Additionally, the dataset used in this study differs considerably from those commonly used in pre-training models (e.g., models trained on COCO). Training from scratch allows the model to better adapt to new task requirements, ensuring that it fully learns relevant features while avoiding irrelevant features or noise introduced by pre-trained weights.

To validate the advantages of the improved models and ensure experimental fairness, both the original and improved models were trained from scratch under identical conditions. Although this approach increases training time and data requirements, it helps accurately evaluate the performance of the improved models and ensures the reliability of the comparative experiments. The specific hyperparameter settings for the model during training are detailed in Table 2.

#### 2.9.1. Implementation Details of the Segmentation Model

The constructed pepper dataset contains 1942 images, divided into training (1624 images), validation (181 images), and test (100 images) sets with a 9:1 ratio. Segmentation experiments are conducted on a Windows server equipped with an NVIDIA Quadro RTX 5000 16 GB GPU, an Intel(R) Core(TM) i9-10900K CPU @ 3.70 GHz, 128 GB RAM, and Windows Server 2019 64-bit OS.

#### 2.9.2. Implementation Details of the Classification Model

The fresh Sichuan peppercorn dataset comprises 4000 images, divided into training (2179 images), validation (234 images), and test (500 images) sets with a 9:1 ratio. Classification experiments are conducted on a Windows computer equipped with an NVIDIA GeForce RTX 2070 SUPER 8 GB GPU, an AMD Ryzen 7 3700X 8-Core Processor CPU @ 3.70 GHz, 16 GB RAM, and Windows 11 Professional 64-bit OS.

### 2.10. Evaluation Metrics

In this experiment, the performance of the models is evaluated using six metrics: parameter count, GFLOPs, accuracy, $mAP_{50}$, recall, and FPS. GFLOPs (giga floating point operations per second) is an important metric for evaluating model complexity. It indicates the speed and computational capability of a neural network model by measuring the number of floating-point operations the model can perform per second.
(10)$$GFLOPs=\frac{XYC_{out}\left(C_{in}\times k\times k+1\right)}{10^{9}}$$
where $C_{in}$ represents the number of input channels, $C_{out}$ represents the number of output channels, k represents the convolution kernel size, X represents the width of the input image, and Y represents the height of the input image.
(11)$$Precision=\frac{TP}{TP+FP}$$
(12)$$Recall=\frac{TP}{TP+FN}$$

Here, Precision indicates the proportion of true positive samples among all samples predicted as positive. Recall indicates the proportion of positive samples correctly identified as positive among all actual positive samples. TP represents the number of true positive cases, FP represents the number of false-positive cases, and FN represents the number of false-negative cases.
(13)$$mAP_{50}=\frac{1}{N}\sum_{x=1}^{N}AP_{x}$$

$mAP_{50}$ (mean average precision) represents the average precision at an IoU threshold of 0.5 for all classes, where N is the number of classes, and $AP_{x}$ is the average precision for class x.
(14)$$FPS=\frac{1}{Inference\;time}$$

To better evaluate the application of the models in smart agriculture scenarios, FPS (frames per second) is included as an evaluation metric in the classification model to assess the real-time inference performance.

In this study, the segmentation model is evaluated using accuracy, recall, $mAP_{50}$, GFLOPs, and parameter count. The classification model is evaluated using accuracy, GFLOPs, parameter count, and FPS.

## 3. Results

In this section, we evaluate the performance of the proposed MultiDomain YOLOv8-seg and MultiDomain YOLOv8-cls models, comparing them with some state-of-the-art (SOTA) instance segmentation and classification models. Ablation studies are also conducted. The segmentation algorithm is trained and validated on the pepper dataset, while the classification algorithm is trained and validated on the fresh Sichuan peppercorn dataset.

Section 3.1 will compare the MultiDomain YOLOv8-seg model with the baseline model and conduct performance comparison experiments with state-of-the-art (SOTA) segmentation models. Section 3.2 will compare the MultiDomain YOLOv8-cls model with the baseline model and perform experiments against SOTA classification models. Section 3.3 will involve ablation experiments. Section 3.4 will validate the high-quality Sichuan pepper selection algorithm.

### 3.1. Validate the Performance of the MultiDomain YOLOv8-seg Model

#### 3.1.1. Comparison of the Segmentation Model with the Baseline Model

This section compares the proposed MultiDomain YOLOv8-seg network with its baseline model YOLOv8-seg [30], evaluating the comprehensive performance of the models in five dimensions: precision, recall, $mAP_{50}$, GFLOPs, and parameter count. Detailed ablation experiments on the MSF3M and MS-DFFM modules will be discussed later. The corresponding experimental results are shown in Table 3. Values in bold indicate the best performance in this comparison.

As shown in Table 3, the proposed MultiDomain YOLOv8-seg network performs best in terms of precision (83.8%), recall (81.4%), and $mAP_{50}$ (88.8%). Its GFLOPs are 20.9 G, only slightly higher than the YOLOv8s-seg model’s 20.7 G. The model size is 5.84 MB, which is 4 MB larger than the YOLOv8n-seg baseline model but much smaller than the YOLOv8s-seg model’s 13.19 MB. The YOLOv8n-seg baseline model shows the lowest performance in precision, recall, and $mAP_{50}$, with values of 77.4%, 77.9%, and 83.4%, respectively, but has the smallest parameter count and GFLOPs at 1.84 MB and 4.6. Overall, the proposed model demonstrates the best performance across all evaluation dimensions, outperforming the YOLOv8s-seg baseline model in precision, recall, $mAP_{50}$, and parameter count.

Figure 14a shows the $mAP_{50}$ training process, while Figure 14b illustrates the loss function decrease process. As depicted, the MultiDomain YOLOv8-seg model ends training early around 150 epochs due to no further improvement in accuracy. It demonstrates superior accuracy, the fastest convergence speed, and the quickest training loss decline among the compared models. 

By incorporating the multi-scale frequency domain feature fusion module (MSF3M) and the multi-scale dual-domain feature fusion module (MS-DFFM) during the feature extraction stage, the model is able to extract and utilize features more efficiently. Based on experimental validation, key hyperparameters such as learning rate, momentum, and weight decay were optimized, as detailed in Table 2. To prevent overfitting and enhance the model’s generalization performance, data augmentation, dropout, and weight regularization techniques were employed during training. Additionally, an early stopping strategy was implemented, which facilitated accelerated convergence in the early stages of training. This approach balanced the model’s training speed and stability, enabling it to reduce loss rapidly and converge to an optimal solution at an earlier stage.

Therefore, the proposed network model outperforms the YOLOv8n-seg and YOLOv8s-seg baseline models in terms of both accuracy and training loss convergence speed.

#### 3.1.2. Comparison with SOTA Segmentation Models

This section compares the MultiDomain YOLOv8-seg with five SOTA segmentation models across six dimensions: precision, recall, $mAP_{50}$, model size, and GFLOPs, to evaluate the proposed model’s performance. The models compared include YOLOv5s-seg, YOLOv7s-seg, YOLOv8s-seg, YOLOv9n-seg, and YOLACT. All experiments were conducted on a Windows Server equipped with an Intel(R) Core(TM) i9-10900K CPU @ 3.70 GHz, 128 GB RAM, and an NVIDIA Quadro RTX 5000 16 GB. The experimental results are shown in Table 4. Values in bold indicate the best performance in this comparison.

As shown in Table 4, YOLOv5s-seg exhibits the highest precision at 85.2% and relatively good recall and $mAP_{50}$ at 81.5% and 87.3%, respectively, but it has a large model size and GFLOPs at 28.3 MB and 25.9. YOLOv7s-seg has slightly lower precision and recall than YOLOv5s-seg at 81% and 81.9%, respectively, but a higher $mAP_{50}$ at 88.2%. Its model size and GFLOPs are also larger, at 36.36 MB and 35.9. YOLOv8s-seg shows relatively low precision and recall at 81% and 81.3%, but it has the smallest model size at only 13.19 MB and low computational complexity at 20.7 GFLOPs. YOLOv9n-seg performs well in recall and $mAP_{50}$, with values of 84.1% and 88.4%, respectively, but has the largest parameter count at 53.35 MB and high computational complexity at 56.1 GFLOPs. YOLACT has the lowest performance across all metrics and the highest computational complexity at 148.2 GFLOPs. The proposed MultiDomain YOLOv8-seg achieves the best $mAP_{50}$ at 88.8%, the smallest model size at 5.84 MB, and low computational complexity at 20.9 GFLOPs. While its precision and recall are slightly lower than YOLOv5s-seg and YOLOv7s-seg, it offers superior overall performance, especially in model size and computational complexity.

As illustrated in Figure 15, the proposed MultiDomain YOLOv8-seg model demonstrates superior overall performance compared to other SOTA segmentation networks in terms of precision, recall, $mAP_{50}$, model size, and GFLOPs. It is more suitable for smart agriculture scenarios and effectively performs the task of Sichuan pepper segmentation.

### 3.2. Validate the Performance of the MultiDomain YOLOv8-cls Model

#### 3.2.1. Comparison of the Classification Model with the Baseline Model

This section compares the proposed MultiDomain YOLOv8-cls network with its baseline model YOLOv8n-cls and evaluates the comprehensive performance of the models across four dimensions: precision, FPS, GFLOPs, and parameter count. The experiments include the addition of the MSF3M and MS-DFFM modules. The experimental results are shown in Table 5. Values in bold indicate the best performance in this comparison.

As shown in Table 5, adding the MSF3M module alone to the YOLOv8n-cls baseline model increases the model size to 0.644 MB and GFLOPs to 0.4 G. However, both precision and FPS decrease to 95.72% and 724 frames, respectively. The MS-DFFM module alone also results in decreased performance, with precision dropping by 2.61% and FPS by 135 frames, while the model size increases by 0.045 MB. When both MSF3M and MS-DFFM modules are added, the network’s precision improves to 98.34%, the highest observed. However, this addition also increases the model size and GFLOPs, resulting in the lowest FPS of 633. This section tests the performance of the proposed MSF3M and MS-DFFM modules, showing that while performance decreases when each module is added separately, it improves when both are added together. This issue will be analyzed in detail in the Section 4.

#### 3.2.2. Comparison with SOTA Classification Models

This section compares the MultiDomain YOLOv8-cls with five SOTA classification models across five dimensions: precision, FPS, model size, and GFLOPs, to evaluate the proposed model’s performance. The models compared include YOLOv5n-cls, Convnext, Efficientnet_b0, Densenet121, and Dpn68. All input images were 640 × 640 × 3, trained for 300 epochs with a batch size of 8. The training ended early if model accuracy did not improve, indicating the model had reached a fitting state. All experiments were conducted on a system with an AMD Ryzen 7 3700X 8-Core Processor CPU @ 3.70 GHz, 16 GB RAM, NVIDIA GeForce RTX 2070 SUPER 8 GB, and a 64-bit Windows 11 Pro operating system. Detailed experimental results are shown in Table 6. Values in bold indicate the best performance in this comparison.

As shown in Table 6, YOLOv5n-cls performs best in FPS at 769 frames due to its smallest model size and GFLOPs of all compared models, at only 0.55 MB and 0.4, respectively. However, it has the lowest precision at 93.82%. The Convnext_tiny network has a precision only slightly lower than the MultiDomain YOLOv8-cls at 98.33%, but its model size and GFLOPs are the largest among the compared models, at 27.8 MB and 36.3, respectively, resulting in poor FPS performance at only 48 frames. Efficientnet_b0 performs well, with a precision of 98.32%, model size and GFLOPs of 4 MB and 3.4 G, and FPS second only to the YOLO series networks at 105 frames. The Densenet121 network has a precision of 97.86%, the lowest FPS at 45 frames, and its parameter count and computational complexity are second only to Convnext_tiny at 6.9 MB and 23.6. The Dpn68 network has intermediate performance among the compared models, with a precision of 98.2%, FPS of 61 frames, and model size and GFLOPs of 11.7 MB and 18.9. The proposed MultiDomain YOLOv8-cls network achieves the highest precision of all models at 98.34%, an FPS of 633 frames, significantly higher than other classification networks, and a model size only 0.098 MB larger than YOLOv5n-cls while improving precision by 4.52%, with the lowest GFLOPs at 0.4.

As shown in Figure 16, the proposed MultiDomain YOLOv8-cls model excels in all four evaluation metrics—precision, FPS, model size, and GFLOPs—compared to other SOTA classification models. It is better suited for the task of classifying semi-mature and mature fresh Sichuan peppers.

### 3.3. Ablation Studies

This section details the ablation experiments for the proposed multi-scale frequency domain feature fusion module (MSF3M) and multi-scale dual-domain feature fusion module (MS-DFFM) and validates the impact of introducing the deep Fourier up-sampling, RepNCSPELAN4, Adown, and C2f-SE modules on network performance. All experiments are based on the YOLOv8n-seg baseline model, as detailed below.

#### 3.3.1. Validation of the MSF3M and MS-DFFM Modules

By adding the MSF3M and MS-DFFM modules, the features at different scales are learned from the spatial, frequency, and channel domains. Compared to the original baseline network, the generated feature maps contain richer information. The MSF3M module, when added to the deeper layers of the network, captures long-distance dependencies more effectively by collecting multi-scale features in the frequency domain. The MS-DFFM module extracts features and information at different scales, maximizing the information content in the images. The specific ablation experiments are shown in Table 7.

As shown in Table 7, adding the MS-DFFM module alone decreases precision by 0.03% compared to the baseline network, but recall is the best at 79.1%, and $mAP_{50}$ increases by 0.07% to 84.1%. The model size and GFLOPs rise to 2.3 MB and 5.0 G, respectively. Adding the MSF3M module alone results in the worst precision performance at 76.7% but the best recall at 78.9%, with $mAP_{50}$ at 83.6%. The model size and GFLOPs are 1.87 MB and 4.7, showing a slight performance improvement over the baseline network. When both MSF3M and MS-DFFM modules are added, the model’s performance significantly improves over the baseline, with precision and $mAP_{50}$ increasing by 1.1% and 1.7%, respectively, while the model size is only 2.32 MB and GFLOPs 5.0. Thus, adding both MSF3M and MS-DFFM modules allows for better feature extraction from the images, with more apparent precision improvements when extracting frequency domain features at deeper network levels.

#### 3.3.2. Validating the Impact of Introducing Modules on Network Performance

This section conducts ablation experiments to validate the impact of different introduced modules on model performance. Table 8 shows the performance impact of various modules. Values in bold indicate the best performance in this comparison.

As shown in Table 8, adding the Adown module reduces the model size to just 1.73 MB and achieves the lowest GFLOPs of 4.6, with precision at 78%, recall at 79.3%, and $mAP_{50}$ at 83.7%, presenting a balanced performance. Adding the C2f-SE module alone slightly reduces the model size by 0.02 MB, with GFLOPs remaining at 4.6, but results in the worst precision and $mAP_{50}$ among all modules at 76.1% and 83.4%, respectively, with Recall at 78.3%. The Fourier up-sampling module performs well in precision and $mAP_{50}$ at 77.8% and 84.1%, respectively, but has the lowest recall at 77.8%, with the model size at 1.96 MB and GFLOPs increased by 0.1 over the baseline model. 

The performance of MSF3M and MS-DFFM modules has been detailed in Section 3.3.1. The RepNCSPELAN4 module significantly improves model precision but adds substantial parameter count and computational complexity, with a model size of 4.43 MB and 18.6 GFLOPs, and precision, recall, and $mAP_{50}$ of 87.1%, 82.8%, and 88.2%, respectively. Overall, the appropriate use of each module is crucial for performance improvement, leading to the design of the MultiDomain YOLOv8-seg network for efficient, accurate, and fast segmentation tasks.

#### 3.3.3. Validating the Impact of the Up-Sampling Operator on the Model

This section analyzes the impact of the up-sampling operator on network performance. As shown in Table 9, replacing the nearest neighbor interpolation up-sampling operator [38] with Fourier up-sampling improves precision and $mAP_{50}$ to 83.8% and 88.8%, respectively. Nearest neighbor interpolation fills new pixel positions with the nearest pixel values, which, despite being efficient, can introduce block artifacts and lose image details. By introducing the deep Fourier up-sampling operator, which effectively separates low and high-frequency components, the up-sampling retains more high-frequency information, producing smoother images with fewer artifacts and better detail preservation. While this method adds 0.04 MB of parameters and keeps GFLOPs at 20.9, it justifies the inclusion of the deep Fourier up-sampling operator by improving network performance.

As shown in Figure 17, the deep Fourier up-sampling operator effectively separates and retains high-frequency information through frequency domain operations, producing smoother feature maps with more detail and fewer artifacts. Nearest neighbor interpolation results in fewer details and blurred edges, leading to stiffer transitions. The deep Fourier up-sampling operator increases model parameters by only 0.004 MB compared to nearest neighbor interpolation, maintaining computational complexity while demonstrating superior performance and validating its inclusion.

#### 3.3.4. Validating of the Impact of Lightweight Modules on the Model

This section evaluates the impact of introducing the C2f-SE and Adown lightweight modules on model performance, validating the rationale for including these modules. Experimental results are shown in Table 10. Values in bold indicate the best performance in this comparison.

As shown in Table 10, adding the Adown module alone significantly reduces the model size to 5.87 MB and GFLOPs to 21.0, with $mAP_{50}$ improving to 88.2%. Adding the C2f-SE module alone does not change the model size or GFLOPs but improves precision, recall, and $mAP_{50}$ to 81.8%, 81.4%, and 88.3%, respectively. Adding both C2f-SE and Adown modules results in significant performance improvements across all metrics, with precision at 83.8%, recall at 81.4%, $mAP_{50}$ at 88.8%, model size at 5.84 MB, and GFLOPs at 20.9. These experiments demonstrate the rationality and importance of adding the C2f-SE and Adown modules for performance improvement.

### 3.4. Validation of Algorithms for Sorting High Quality Fresh Peppers

This section validates the proposed algorithms for sorting high-quality fresh Sichuan peppers. Images are first input into the MultiDomain YOLOv8-cls for sorting mature peppers. Then, the screened 678 images undergo average local pixel value difference and flat area marking. Finally, the generated binary images are evaluated for sparsity and black pixel proportion to output the sorting results, selecting high-quality Sichuan pepper grains. The corresponding prediction results are shown in Figure 18.

## 4. Discussion

### 4.1. Analysis of the Impact of MSF3M and MS-DFFM Modules on Network Performance

This section will focus on analyzing why the individual addition of the multi-scale frequency domain feature fusion module (MSF3M) and the multi-scale dual-domain feature fusion module (MS-DFFM) to a classification network leads to a decline in model performance and explore the reasons behind the significant improvement in performance when both MSF3M and MS-DFFM modules are included simultaneously.

#### 4.1.1. Why Adding MSF3M and MS-DFFM Modules Separately Leads to a Decline in Network Performance

This section will specifically examine why adding either the MSF3M or MS-DFFM module individually leads to a decline in model performance within a classification network. For the dataset of fresh Sichuan peppercorns constructed in this study, the feature extraction process focuses more on local details, textures, and color features of the peppercorns, with relatively less emphasis on global information. The classification model network has a relatively shallow depth, while the MS-DFFM module primarily extracts multi-scale information from the images. The use of large-scale convolution introduces more global feature information, which, although beneficial for local feature extraction, can also introduce additional noise or lead to overfitting, resulting in a slight decrease in model performance. On the other hand, the MSF3M module mainly performs global feature extraction in the frequency domain, with relatively less emphasis on local information. The lack of these detailed local features can lead to a decline in model accuracy.

#### 4.1.2. Why Adding Both MSF3M and MS-DFFM Modules Simultaneously Improves Network Performance

This section will analyze why the simultaneous addition of MSF3M and MS-DFFM modules leads to a significant improvement in model performance. The introduction of both modules provides complementary feature extraction methods. MSF3M extracts global information in the frequency domain, compensating for the lack of global feature information in the spatial domain. On the other hand, the MS-DFFM module focuses on local spatial and channel features, capturing finer-grained details. By complementing global and local feature information, the model greatly enriches the image features, enhances feature representation capabilities, reduces the risk of overfitting, and gains a more comprehensive understanding of the input data, leading to a substantial improvement in network performance. The generated features are illustrated in Figure 19.

As shown in Figure 19, after incorporating both the MSF3M and MS-DFFM modules, the feature maps exhibit a more balanced distribution. This approach retains the detail-capturing ability of the MSF3M module while enhancing the salient feature extraction capability of the MS-DFFM module. Consequently, the simultaneous use of these two modules significantly enhances the network’s feature extraction capacity.

Figure 20 presents the results of visual feature extraction using Grad-CAM for the proposed MultiDomain YOLOv8 model and the YOLOv8 model. After feature extraction on Sichuan pepper, mature peppercorns, and semi-mature peppercorns through the MS-DFFM and MSF3M modules, the generated heatmaps demonstrate a more refined feature extraction capability, better separating pepper clusters from the background. The visualization shows clearer contours and significant color variations, with stronger contrast in the highlighted areas, which more effectively highlights the key features of the peppercorns and reduces noise interference. Compared to the baseline network, the MultiDomain YOLOv8 model more clearly captures the texture details and differences in various regions of the peppercorns. Experimental results indicate that the MS-DFFM and MSF3M modules proposed in this study significantly improve feature extraction accuracy, background separation, response to different maturity levels, and anti-interference capability in the tasks of Sichuan pepper segmentation and maturity classification, demonstrating their effectiveness and reliability compared to the YOLOv8 model.

### 4.2. Limitations of the Quality Sorting Algorithm for Fresh Sichuan Pepper

In practical agricultural applications, various factors can influence the high-quality sorting of Sichuan peppercorns. The following analysis examines how these factors affect the algorithm:
Lighting Effects: In real sorting conditions, variations in lighting can significantly impact the quality of peppercorn images. Especially in outdoor environments, lighting conditions change with time, weather, and location, leading to noticeable variations in image brightness and contrast. Such inconsistencies can cause a decrease in model performance during the training and testing phases of deep learning algorithms. In this study, the model was trained on images captured under fixed lighting conditions. However, in practical scenarios with different lighting conditions, the model’s recognition accuracy may be compromised. Moreover, shadows can cause anomalies in local pixel values, leading to errors in methods like average local pixel value difference and flat region marking, which may result in more incorrect flat region labels, affecting the algorithm’s decision-making.Impact of Classification Algorithm: When classifying mature and semi-mature peppercorns, misclassification may occur, where some semi-mature peppercorns are incorrectly categorized as mature. Semi-mature peppercorns may have a mix of red and green colors, and if both colors are uniformly distributed, the average local pixel value difference method might mark them as flat regions, thus influencing the algorithm’s judgment.Image Acquisition Impact: During the actual image acquisition process, issues such as occlusion between peppercorn clusters and the angle of shooting can result in incomplete assessments of peppercorn quality. Although this algorithm captures images from both the front and back of individual peppercorn clusters, blind spots may still exist. Occlusion may prevent some semi-mature peppercorns from being captured in the image, and subsequent cropping may fail to include all the peppercorns in the cluster, thereby affecting the algorithm’s ability to accurately assess the quality of the peppercorn cluster.Impact of Image Noise: When collecting peppercorn images under uncontrolled conditions, the images may contain various types of noise, such as environmental noise, sensor noise, and motion noise. These noises can hinder the model’s ability to extract key features from the images, leading to a decrease in the accuracy of classification and segmentation algorithms. Especially when the signal-to-noise ratio is low, the algorithm may struggle to distinguish between the target and the background, making feature extraction more challenging and resulting in reduced model performance.


### 4.3. Future Prospects and Agricultural Applications

The MultiDomain-YOLOv8 model, which integrates multi-domain and multi-scale features, has demonstrated outstanding performance in Sichuan pepper segmentation and grading tasks by more comprehensively and accurately extracting image features. These advantages suggest that this algorithm holds significant potential for broader agricultural applications beyond Sichuan pepper grading. Specific potential applications include:

Crop Health Detection: The algorithm’s ability to integrate multi-scale features enables precise identification of various stages of crop diseases and pests, such as leaf spot detection and pest detection on leaves. By combining multi-domain image features, the algorithm can better capture subtle differences in diseased areas. This capability allows for timely intervention, helping farmers to control pests and diseases effectively, thereby reducing losses and ensuring the healthy growth of crops.

Fruit Ripeness Grading: Beyond Sichuan pepper, the MultiDomain-YOLOv8 model can be applied to ripeness detection of other fruits, such as apples, citrus, and grapes. Different stages of fruit ripeness exhibit significant differences in color, texture, and shape. The integration of multi-domain and multi-scale features enhances the model’s ability to identify detailed characteristics, accurately distinguishing between ripe and unripe fruits, thereby improving sorting and grading efficiency within the agricultural supply chain.

Applications in Smart Agricultural Machinery: The strong feature extraction capability of MultiDomain-YOLOv8 can provide precise visual guidance for automated harvesting robots. During the harvesting process, the model can accurately identify ripe fruits or monitor crop growth, facilitating intelligent harvesting and management.

In summary, the multi-scale and multi-domain feature extraction capabilities of the MultiDomain-YOLOv8 model are expected to further expand its applications in various agricultural domains, contributing to the development of smart agriculture.

## 5. Conclusions

This study proposes the MultiDomain YOLOv8 model for the instance segmentation and classification of Sichuan pepper, along with a method for selecting high-quality Sichuan pepper. Extensive experiments demonstrate the superiority of the proposed algorithm. The study introduces the multi-scale frequency domain feature fusion module (MSF3M) and the multi-scale dual-domain feature fusion module (MS-DFFM), focusing on multi-domain and multi-scale feature extraction to enhance network performance. Additionally, modules such as Adown, C2f-SE, Fourier up-sampling, and RepNCSPELAN4 are incorporated to address issues related to high computational complexity and sampling inaccuracies in certain modules, further accelerating network inference and improving accuracy. The scientific validity and rationality of introducing these modules are confirmed through extensive experimentation.

Compared to the YOLOv8s-seg baseline network, the proposed MultiDomain YOLOv8-seg network shows an increase of 2.8% in accuracy and 1.3% in $mAP_{50}$, with a reduction of 7.35 MB in parameters and GFLOPs of only 20.9. The MultiDomain YOLOv8-cls network achieves an accuracy of 98.34%, an FPS of 633 frames, and a model size of only 0.648 MB, outperforming the compared state-of-the-art models in various aspects. The proposed model makes significant advancements in reducing model size and computational requirements, making it more suitable for real-time applications. These improvements are expected to benefit the Sichuan pepper industry by enabling more precise and faster sorting operations, further supporting the development of agricultural sorting robots for Sichuan pepper.

In addressing the computational resource constraints in smart agriculture scenarios due to cost issues, the proposed network model, with its low parameter count and fast computation speed, is more advantageous for the design and deployment of sorting robots and network models for Sichuan pepper. Future work will focus on the practical application of the proposed algorithm in sorting robots, with ongoing improvements to address any arising challenges, ultimately achieving more accurate and rapid sorting and providing a reference solution for sorting high-quality Sichuan pepper.

## Figures and Tables

**Figure 1 foods-13-02776-f001:**
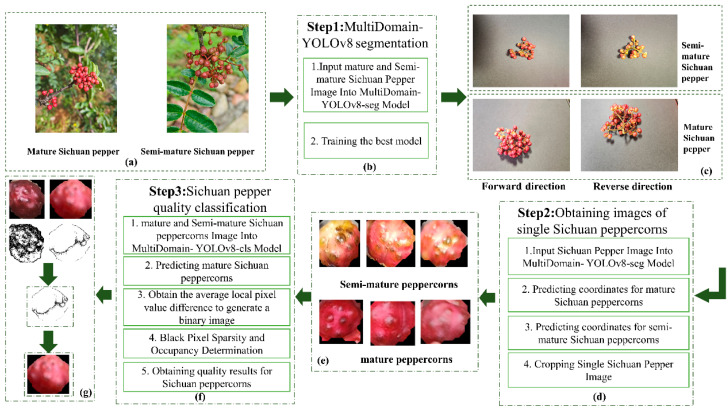
A quality classification system diagram for fresh Sichuan pepper. (**a**) Sichuan pepper sample images. (**b**) Sichuan pepper image feature extraction process. (**c**) Sichuan pepper front and back image acquisition. (**d**) Single Sichuan pepper image acquisition process. (**e**) Single Sichuan pepper image (**f**) High-quality Sichuan pepper sorting process. (**g**) High-quality Sichuan pepper sorting images.

**Figure 2 foods-13-02776-f002:**
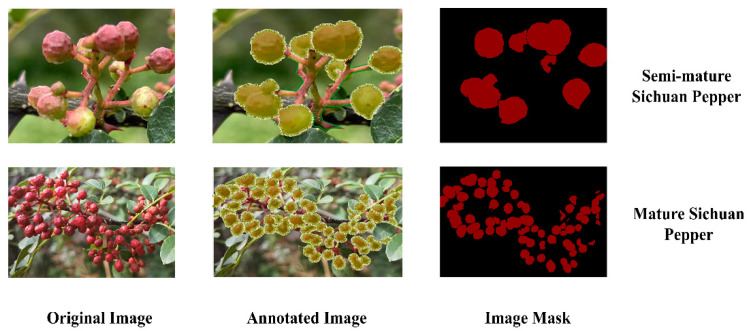
Original and annotated images of Sichuan pepper.

**Figure 3 foods-13-02776-f003:**
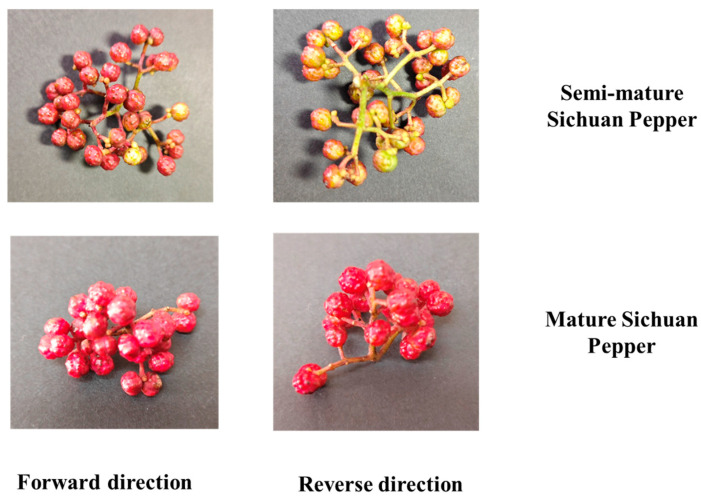
Collected images of front and back views of fresh Sichuan pepper clusters.

**Figure 4 foods-13-02776-f004:**
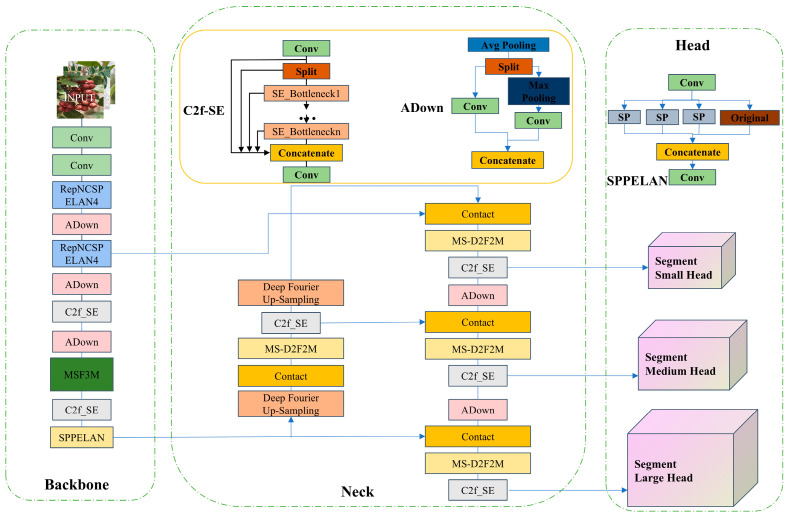
MultiDomain YOLOv8-seg network architecture.

**Figure 5 foods-13-02776-f005:**
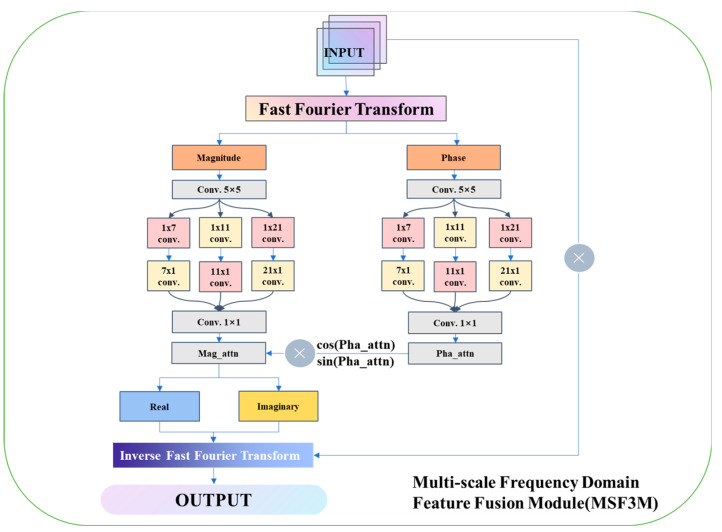
MSF3M architecture diagram.

**Figure 6 foods-13-02776-f006:**
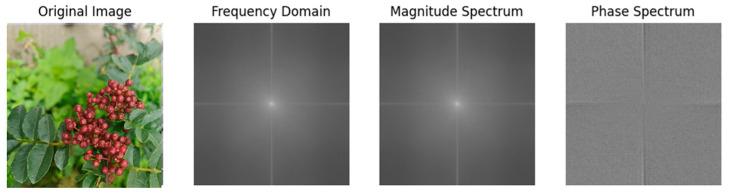
Magnitude and phase images transformed from the spatial domain to the frequency domain.

**Figure 7 foods-13-02776-f007:**
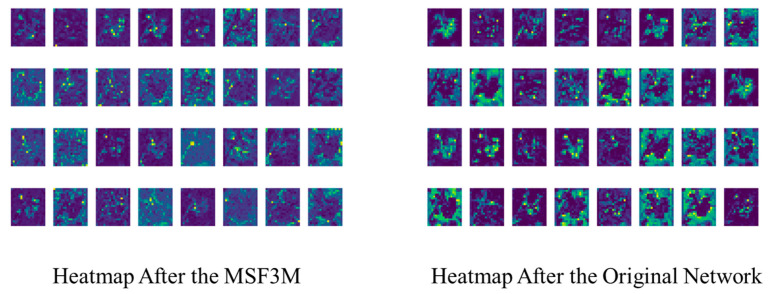
Figure shows the feature maps processed with and without the MSF3M module.

**Figure 8 foods-13-02776-f008:**
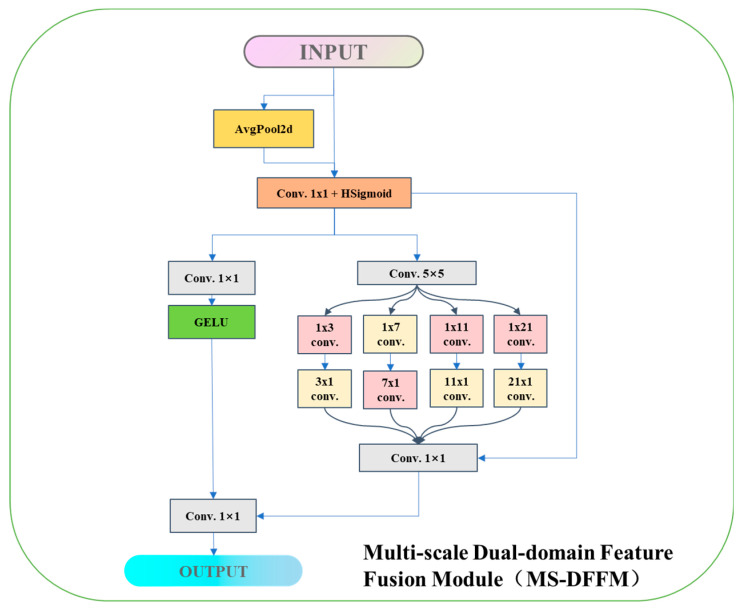
MS-DFFM architecture diagram.

**Figure 9 foods-13-02776-f009:**
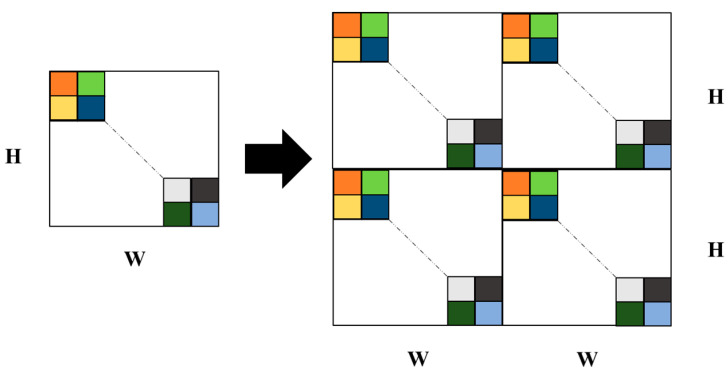
Padding process in the deep Fourier up-sampling module.

**Figure 10 foods-13-02776-f010:**
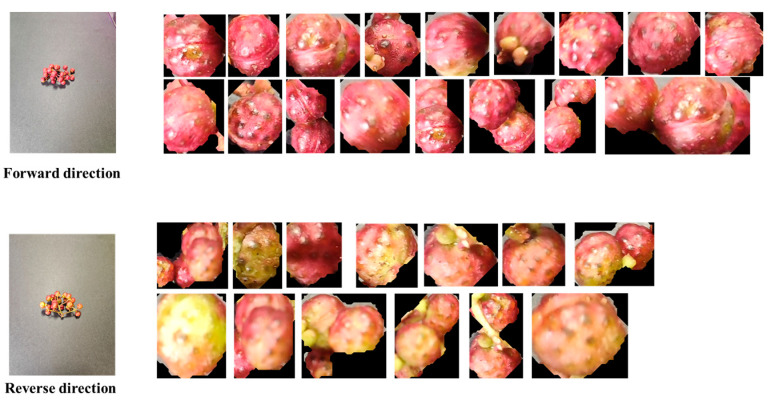
Dataset images of mature and semi-mature Sichuan peppers.

**Figure 11 foods-13-02776-f011:**
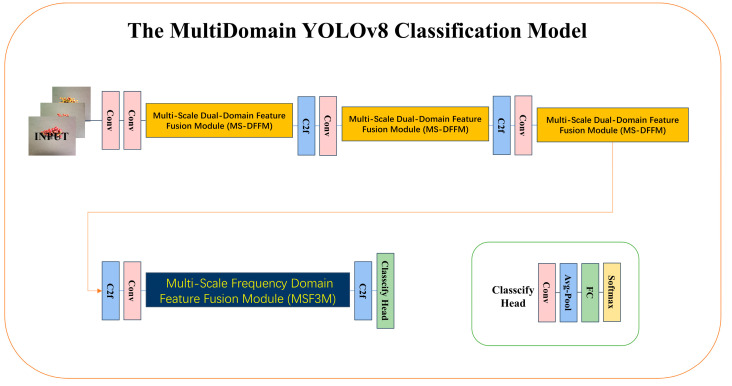
MultiDomain YOLOv8-cls network architecture.

**Figure 12 foods-13-02776-f012:**
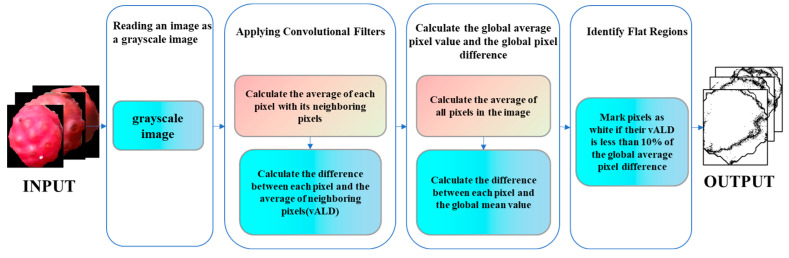
Flow of the average local pixel value difference method.

**Figure 13 foods-13-02776-f013:**
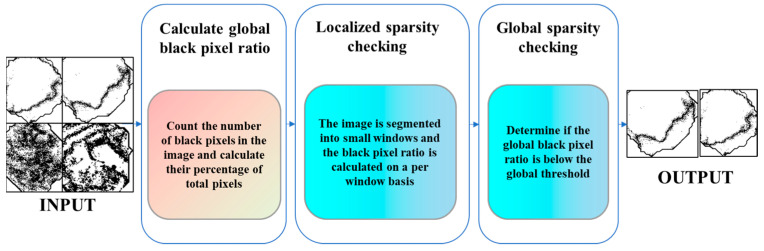
Flow of the flat region labeled image assessment.

**Figure 14 foods-13-02776-f014:**
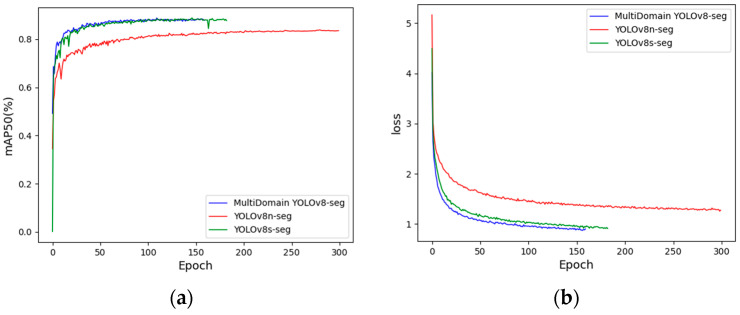
Convergence process comparison of accuracy and loss function.

**Figure 15 foods-13-02776-f015:**
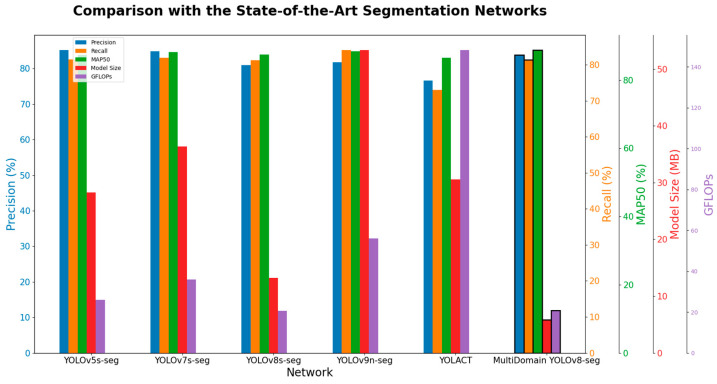
Performance comparison with SOTA segmentation models.

**Figure 16 foods-13-02776-f016:**
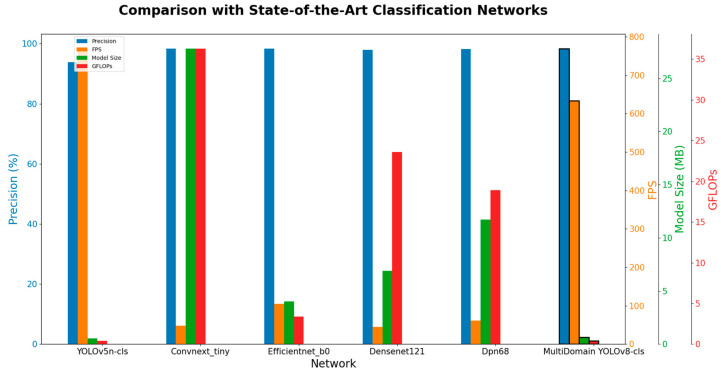
Performance comparison with SOTA classification models.

**Figure 17 foods-13-02776-f017:**
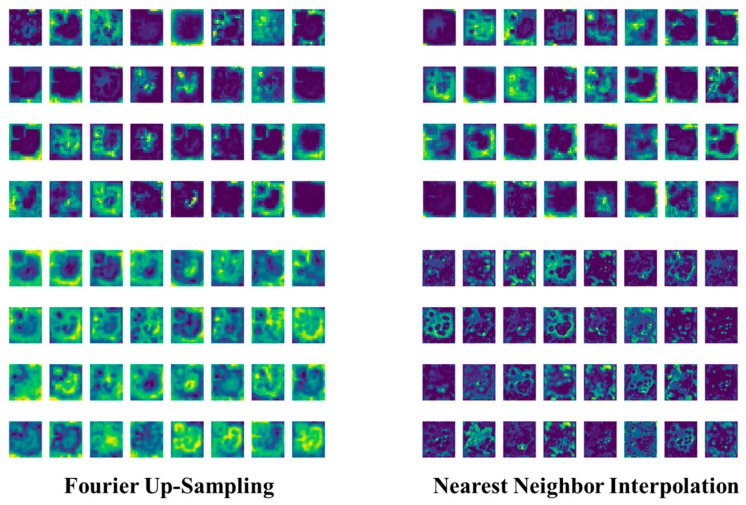
Feature maps of deep Fourier up-sampling and nearest neighbor interpolation up-sampling operators.

**Figure 18 foods-13-02776-f018:**
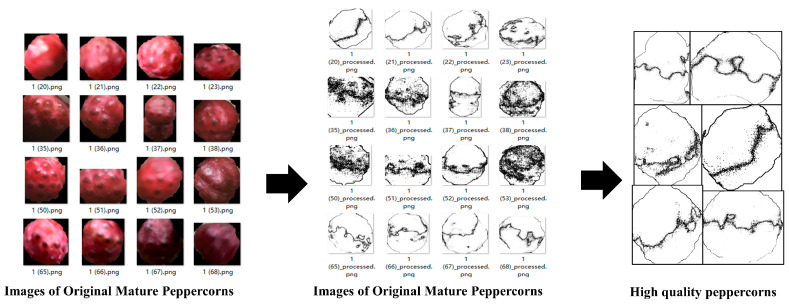
Prediction results of high-quality fresh pepper sorting algorithm.

**Figure 19 foods-13-02776-f019:**
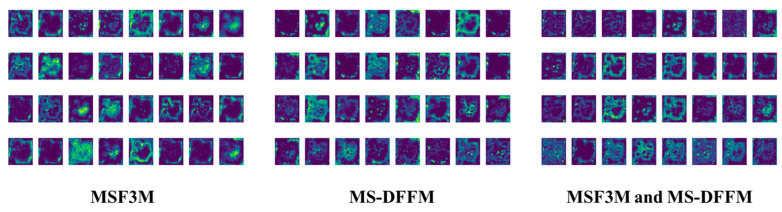
Feature maps generated by MSF3M and MS-DFFM.

**Figure 20 foods-13-02776-f020:**
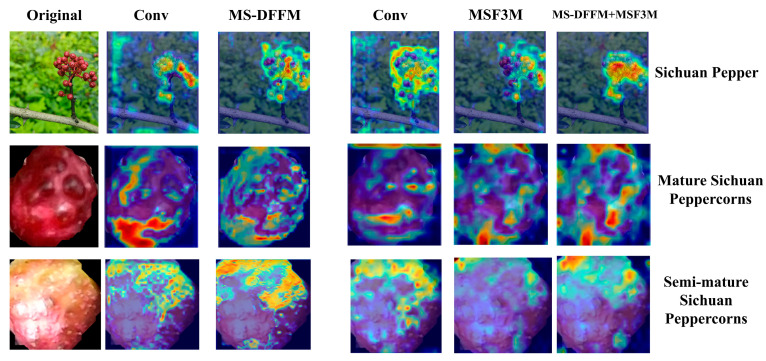
Grad-CAM visualization of MultiDomain YOLOv8 and YOLOv8 models.

**Table 1 foods-13-02776-t001:** Quality grading standards for fresh Sichuan pepper.

Rank	Grading Criteria
First class	Color: Bright red or reddish Aroma: Fragrant, aromatic, without off-flavors Impurities: No thorns, no moldy or rotten grains, fruit clusters with 1–2 compound leaves, few stems, and no mixed peppercorns
Second class	Color: Uneven coloring, mixed shades. Aroma: Mildly fragrant with a subtle aroma, free from any unusual smell. Impurities: Free from thorns or moldy grains, featuring panicles and compound leaves, without foreign peppercorns.

**Table 2 foods-13-02776-t002:** Model training hyperparameters.

Hyper-Parameter	Value
Epoch	300
BatchSize	16
ImageSize	640 × 640 × 3
Optimizer	SGD
Patience	50
Pretrained	False
Weight_decay	0.0005
Workers	4
lr0	0.01
lrf	0.01
Momentum	0.937

**Table 3 foods-13-02776-t003:** Comparison of MultiDomain YOLOv8-seg and baseline networks.

Model	Precision (%)	Recall (%)	$mAP_{50}$ (%)	Model Size (MB)	GFLOPs (G)
YOLOv8n-seg	77.4	77.9	83.4	**1.84**	**4.6**
YOLOv8s-seg	81.0	81.3	87.5	13.19	20.7
MultiDomain YOLOv8-seg	**83.8**	**81.4**	**88.8**	5.84	20.9

**Table 4 foods-13-02776-t004:** Comparison of MultiDomain YOLOv8-seg with SOTA segmentation models.

Model	Precision (%)	Recall (%)	$mAP_{50}$ (%)	Model Size (MB)	GFLOPs (G)
YOLOv5s-seg [31]	85.2	81.5	87.3	28.3	25.9
YOLOv7s-seg [32]	84.8	81.9	88.2	36.36	35.9
YOLOv8s-seg [30]	81.0	81.3	87.5	13.19	20.7
YOLOv9n-seg [23]	81.8	84.1	88.4	53.35	56.1
YOLACT [33]	76.6	73.0	86.5	30.56	148.2
MultiDomain YOLOv8-seg	83.8	81.4	**88.8**	5.84	20.9

**Table 5 foods-13-02776-t005:** Comparison of MultiDomain YOLOv8-cls and baseline networks.

MSF3M	MS-DFFM	Precision (%)	FPS	Model Size (MB)	GFLOPs (G)
×	×	96.31	**838**	**0.601**	**0.3**
×	√	95.72	724	0.644	0.4
√	×	93.70	703	0.646	0.4
√	√	**98.34**	633	0.648	0.4

**Table 6 foods-13-02776-t006:** Comparison of MultiDomain YOLOv8-cls with SOTA classification models.

Model	Precision (%)	FPS	Model Size (MB)	GFLOPs (G)
YOLOv5n-cls [31]	93.82	**769**	**0.55**	**0.4**
Convnext_tiny [34]	98.33	48	27.8	36.3
Efficientnet_b0 [35]	98.32	105	4.0	3.4
Densenet121 [36]	97.86	45	6.9	23.6
Dpn68 [37]	98.2	61	11.7	18.9
MultiDomain YOLOv8-cls	**98.34**	633	0.648	**0.4**

**Table 7 foods-13-02776-t007:** Performance validation of MSF3M and MS-DFFM modules.

MSF3M	MS-DFFM	Precision (%)	Recall (%)	$mAP_{50}$ (%)	Model Size (MB)	GFLOPs (G)
×	×	77.4	77.9	83.4	1.84	4.6
×	√	77.1	79.1	84.1	2.3	5.0
√	×	76.7	78.9	83.6	1.87	4.7
√	√	78.5	77.2	85.1	2.32	5.0

**Table 8 foods-13-02776-t008:** Performance impact of introduced modules.

Adding Modules	Precision (%)	Recall (%)	$mAP_{50}$ (%)	Model Size (MB)	GFLOPs(G)
Adown	78.0	79.3	83.7	**1.73**	**4.6**
C2f-SE	76.1	78.3	83.4	1.82	**4.6**
Fourier Up-Sampling	77.8	77.8	84.1	1.96	4.7
MSF3M	76.7	78.9	83.6	1.87	4.7
MS-DFFM	77.1	79.1	84.1	2.30	5.0
RepNCSPELAN4	**87.1**	**82.8**	88.2	4.43	18.6

**Table 9 foods-13-02776-t009:** Comparison of deep Fourier up-sampling and nearest neighbor interpolation up-sampling operators.

Up-Sampling	Precision (%)	Recall (%)	$mAP_{50}$ (%)	Model Size (MB)	GFLOPs (G)
Fourier up-sampling	83.8	81.4	88.8	5.84	20.9
Nearest neighbor interpolation	82.6	82.1	88.4	5.80	20.9

**Table 10 foods-13-02776-t010:** Impact of C2f-SE and Adown modules on network performance.

Lightweight Modules	Precision (%)	Recall (%)	$mAP_{50}$ (%)	Model Size (MB)	GFLOPs (G)
×	81.2	81.3	87.9	6.02	21.1
Adown	80.4	81.3	88.2	5.87	21.0
C2f-SE	81.8	81.4	88.3	6.02	21.1
C2f-SE+ Adown	**83.8**	**81.4**	**88.8**	**5.84**	**20.9**

## Data Availability

The original contributions presented in the study are included in the article, further inquiries can be directed to the corresponding author.

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
