# Peer review of "A Method for Sorting High-Quality Fresh Sichuan Pepper Based on a Multi-Domain Multi-Scale Feature Fusion Algorithm"

_foods, 2024, doi:10.3390/foods13172776_

Round 1

Reviewer 1 Report

Comments and Suggestions for Authors

REVIEW COMMENTS

foods-3169072

Title:    A Method for Sorting High-Quality Fresh Sichuan Pepper Based on a Multi-Domain Multi-Scale Feature Fusion Algorithm.”

1.      This study introduces innovative Multi-Domain YOLOv8 models for efficient, high accuracy sorting of Sichuan pepper, showcasing impressive segmentation and classification results. The proposed methods significantly enhance the speed and precision of sorting, proving well-suited for deployment on mobile devices.

2.      The introduction effectively sets the stage for the study by outlining the significance of Sichuan pepper and the need for advanced grading techniques. To enhance clarity and focus, consider streamlining the discussion on emphasizing the innovative aspects of your proposed method.

3.      In section 2, Line 128-129, The Figure-1, pictures along with the steps, It is suggested to show the steps in the text and denote with captions for each step in the figure inside so that the legibility of the letters could be clear enough for the readers.

4.      In line 144-145 “The collected Sichuan pepper images were annotated using the anylabeling tool integrated with the SAM model”………..the authors have mentioned that “anylabeling tool” which is very vague and it is suggested to mention specific labelling tools that have been used.

5.      There is no much difference between “Original and Annotated Images” in figure2

6.      It is not mentioned in the manuscript why the authors have chosen multi-domain YOLOv8 network model for the proposed study when there are many other similar network models available in the literature.

7.      In section3.2 Evaluation metrics, line-479 the expansion must be given for “GFLOP” for the first time usage in the manuscript. It is also suggested to mention the reference and citation for all the performance metrics formulae used in the proposed study.

8.      The authors have never mentioned the term “Multi-Domain Multi-Scale Feature Fusion Algorithm” other than the title. Then what is the reason for mentioning it in the title.

9.      The words mentioned in Figure 15, Figure 16 have to be legible for the readers.

10.  In section 4, The authors have given explanation, but it could benefit from a more concise explanation of the specific reasons behind the performance decline when modules are added individually. Emphasizing how the complementary nature of these modules improves overall performance could further clarify the significance of their combined use.

11.  The conclusion effectively summarizes the study's contributions and results. It could be improved by providing a succinct overview of the main findings and their implications with a practical impact of the reduced model size and computational demands on real time applications that would strengthen the conclusion part.

Reviewer 2 Report

Comments and Suggestions for Authors

The manuscript “A Method for Sorting High-Quality Fresh Sichuan Pepper Based on a Multi-Domain Multi-Scale Feature Fusion Algorithm" (foods-3169072) requires several improvements before it can be recommended. Therefore, I am suggesting this work for major revisions.

As major observations, which must be attended to, I highlight:

1. I noticed some grammatical errors in the writing; therefore, I suggest having the English reviewed by a native speaker.

2. The authors must reformulate the abstract. Please note that you are presenting an abstract with 228 words, while Foods limits abstracts to 200 words. I also highlight that the authors must follow the premise of presenting the highlights of the results in the abstract, something I did not observe in this summary.

3. Lines 25-27: The statement that "MultiDomain YOLOv8-cls achieves a classification accuracy of 98.34%" needs to be justified with clearer context, explaining why this accuracy is significant compared to other existing approaches. I suggest adding a sentence that compares the results with other approaches or explains the practical relevance of this accuracy in an agricultural context.

4. Lines 33-36: The introduction mentions that the quality of Sichuan pepper varies due to environmental factors, but specific examples of these factors and how they affect quality grading are missing. It would be interesting to include studies or data showing how specific climatic variations impact the ripening and quality of the peppers.

5. Lines 39-40: The sentence "Grading fresh Sichuan pepper can also improve the efficiency and quality of subsequent processing" could be more detailed. Add information about the economic impact of this efficiency improvement, such as potential post-harvest loss reduction or increased profitability for producers.

6. Lines 136-140: When describing the image collection process, details about lighting conditions and camera positioning are missing. These variables can significantly impact the segmentation model's results. I suggest adding information about the exact camera settings and how the lighting conditions were controlled to ensure image consistency.

7. Lines 152-156: The collection of images from both sides of the pepper clusters is mentioned, but the impact of this collection on the model’s training and final results is unclear. Explain why this approach was adopted and how it contributed to the model’s accuracy.

8. Lines 462-468: The text mentions that the models were trained from scratch, without using pre-trained weights. It would be interesting to explain why this choice was made, especially compared to using pre-trained weights, which often speed up training and improve initial accuracy. Discuss the implications of this choice on training time and data requirements.

9. Lines 520-524: The comparison of convergence processes between the different models could be expanded to include a more detailed analysis of what contributed to the superior performance of MultiDomain YOLOv8-seg. The sentence "demonstrates superior accuracy, the fastest convergence speed, and the quickest training loss decline" could be complemented with specific explanations about hyperparameter tuning, architecture, or regularization techniques that may have contributed to these results.

10. Lines 557-572: The analysis of the addition of the MSF3M and MS-DFFM modules to the YOLOv8-cls model could be further developed. Clarify which specific features of these modules contribute to the accuracy increase. If possible, include a qualitative comparison of the feature maps generated with and without these modules to visually illustrate how the modules improved feature extraction.

11. Although the limitations of the proposed algorithm are briefly mentioned, the discussion could be more specific. For example, discuss the challenges encountered in real-world application scenarios, such as variations in natural lighting in the field or the presence of noise in the images. How might these limitations impact the model's performance in uncontrolled conditions?

12. In the conclusion, while the work is adequately summarized, it would be helpful to add a final section discussing future applications of the algorithm in other areas of agriculture beyond Sichuan pepper grading. This would provide a broader perspective on the work's potential and encourage the exploration of new applications.

As a minor and main note, I highlight:

1. The section "3.1" and its subsections do not seem like Results, but rather a methodological description. Therefore, relocate this section to Materials and Methods.

2. Please increase the font size of the information in Figures 15 and 16.

3. All figures and tables should have a 12-point spacing before and a 6-point spacing after.

4. Section "4.1.1.": Justify the text and apply a 12-point spacing before and a 3-point spacing after. Note, review all sections and subsections of the work, as the authors make this error at various points.

Comments on the Quality of English Language

Moderate editing of English language required.

Round 2

Reviewer 2 Report

Comments and Suggestions for Authors

Considering the corrections provided, I am considering this work for publication. I would like to take this opportunity to congratulate the authors for their commitment and dedication to substantial improvements in this study.